# RNA and Protein Determinants Mediate Differential Binding of miRNAs by a Viral Suppressor of RNA Silencing Thus Modulating Antiviral Immune Responses in Plants

**DOI:** 10.3390/ijms23094977

**Published:** 2022-04-29

**Authors:** Robert Pertermann, Ralph Peter Golbik, Selvaraj Tamilarasan, Torsten Gursinsky, Selma Gago-Zachert, Vitantonio Pantaleo, Iris Thondorf, Sven-Erik Behrens

**Affiliations:** 1Institute of Biochemistry and Biotechnology, Martin Luther University Halle-Wittenberg, Saale, 06120 Halle, Germany; robert.pertermann@t-online.de (R.P.); ralph.golbik@biochemtech.uni-halle.de (R.P.G.); s.tamilarasan@verovaccines.org (S.T.); torsten.gursinsky@biochemtech.uni-halle.de (T.G.); selma.gago-zachert@bct.uni-halle.de (S.G.-Z.); iris.thondorf@biochemtech.uni-halle.de (I.T.); 2Department of Biology, Agricultural and Food Sciences, Institute for Sustainable Plant Protection, Bari Unit, CNR, 70126 Bari, Italy; vitantonio.pantaleo@cnr.it

**Keywords:** RNA silencing, suppressor, G–U wobble, RNA virus, plant, RNA–protein interaction

## Abstract

Many plant viruses express suppressor proteins (VSRs) that can inhibit RNA silencing, a central component of antiviral plant immunity. The most common activity of VSRs is the high-affinity binding of virus-derived siRNAs and thus their sequestration from the silencing process. Since siRNAs share large homologies with miRNAs, VSRs like the *Tombusvirus* p19 may also bind miRNAs and in this way modulate cellular gene expression at the post-transcriptional level. Interestingly, the binding affinity of p19 varies considerably between different miRNAs, and the molecular determinants affecting this property have not yet been adequately characterized. Addressing this, we analyzed the binding of p19 to the miRNAs 162 and 168, which regulate the expression of the important RNA silencing constituents Dicer-like 1 (DCL1) and Argonaute 1 (AGO1), respectively. p19 binds miRNA162 with similar high affinity as siRNA, whereas the affinity for miRNA168 is significantly lower. We show that specific molecular features, such as mismatches and ‘G–U wobbles’ on the RNA side and defined amino acid residues on the VSR side, mediate this property. Our observations highlight the remarkable adaptation of VSR binding affinities to achieve differential effects on host miRNA activities. Moreover, they show that even minimal changes, i.e., a single base pair in a miRNA duplex, can have significant effects on the efficiency of the plant antiviral immune response.

## 1. Introduction

In plants, the RNA silencing process is a crucial component of the immune response against pathogens such as viruses. Silencing is induced by double stranded (ds) elements of viral RNA genomes and/or mRNAs, dsRNA viral replication intermediates or dsRNAs that are generated from viral RNA templates by host RNA-dependent RNA polymerases (RDRs) [1,2]. The dsRNA elements are mainly detected by the Dicer-like proteins DCL4 or DCL2, which process them into 21 and 22 nt viral small interfering RNA (vsiRNA) duplexes, respectively [3,4]. Accordingly, during an infection, vsiRNAs accumulate and may be further amplified by the activity of RDRs [1,2,5,6,7,8,9,10]. Argonaute (AGO) endonucleases [11], which are the active components of RNA-induced silencing complexes (RISC), incorporate the ‘guide strand’ of an siRNA duplex while the other, ‘passenger strand’ is removed [12]. Mainly via base-pairing, the guide strand directs the RISC to the target RNA, which commonly is the cognate viral RNA [13,14]. The well-characterized antiviral AGO1 and AGO2 proteins then target these RNAs predominantly by endonucleolytic cleavage (‘slicing’) in a vsiRNA-directed, sequence-specific manner [15,16,17,18,19]. Thus, the spread of primary and secondary (RDR-generated) vsiRNAs can lead to a reduction in viral titer and induction of local and systemic plant immunity, respectively [8,14].

Most plant viruses encode one or more proteins that function as viral suppressors of RNA silencing, VSRs, and can block antiviral RNA silencing during different stages [20]. One of the best-characterized VSRs is the p19 protein of Tombusviruses (family *Tombusviridae*). p19 binds siRNA duplexes with high affinity, and siRNA binding was found to depend on the size but not on the sequence of the RNA molecule [20,21,22,23,24,25]. The p19-mediated sequestration of siRNAs prevents RISC assembly [21] and interferes with the systemic spread of silencing [22,23,24,25].

The other important function of RNA silencing is the regulation of plant gene expression at the post-transcriptional level. During this process, RISCs form with endogenous microRNAs (miRNAs, miRs) and silence cellular mRNA targets. The Dicer-like protein DCL1 processes miRNA duplexes from cellular pri-miRNA transcripts [26], and AGO1 is the main catalytic component of miRNA-containing RISC. After removal of the miRNA passenger strand (also referred to as the ‘star strand’), AGO1/RISC associate with the target RNA through base pairing of the incorporated guide strand, and silence the RNA either by slicing or by inhibiting translation [26,27,28,29].

In dicots, two miRNAs play a central role in regulating RNA silencing: miR162 targets the DCL1 mRNA [30] and miR168 targets the AGO1 mRNA [31,32,33,34]. We, and others, have shown that the p19 VSR binds miR162 considerably more effectively than miR168 [35,36]. Moreover, we could recently demonstrate that the differential binding of the miRNAs modulates the expression of DCL1 and AGO1 in the early stage of a viral infection [36].

Here, we characterized the molecular determinants of the differential binding of miR162 and miR168 by p19. We show that mismatches and G–U wobbles determine the protein–RNA interactions on the side of the miRNA, while a defined stretch of amino acids regulates the RNA affinity of p19 on the protein’s side. Accordingly, our data suggest that even small changes in both the cellular miRNA and/or viral protein can have significant effects on the regulation of the plant antiviral immune response.

## 2. Results

Nucleotide mismatches and ‘G–U wobbles’ determine the affinities of miRNAs to p19. In the following experiments, we mainly used the purified p19 of *Carnation Italian ringspot virus*, *CIRV*, unless otherwise described. Following a previously established procedure, we expressed the protein in *E. coli* and purified it in two chromatographic steps [36] (see also Section 4). The resulting p19 was free of nucleic acids contaminants and confirmed to form protein dimers [36] (data not shown). To measure RNA binding, we applied an electrophoretic mobility shift assay (EMSA) with synthetic miRNAs according to the protocol of Vargason et al. [22]; a summary of all measured binding data is given in Table 1. First, we confirmed earlier data [32,36,37,38] by showing that p19 has a dissimilar binding behavior with different miRNAs: miR162 (miR162a from *Arabidopsis thaliana*, *At*, was used) exhibits high siRNA-like affinity, whereas miR168 (miR168a from *At*, was used), in contrast, shows much weaker binding to p19 (Figure 1).

To define the molecular features that determine the strongly divergent substrate preferences of p19 for the miRNAs 162 and 168, we modified the ‘high-affinity binder’ *At*miR162 sequentially. For this, we took advantage of the fact that p19 exists and functions as a dimer: the p19 dimer binds an siRNA such that the monomer-subunits symmetrically associate with the respective termini of the sRNA duplex, with each covering roughly 10 nucleotides/one turn of the dsRNA A helix [22,23] (see also the following text). To evaluate the role of individual nucleotides for binding, we modified only one terminus of the *At*miR162. Note that in the text and figures, the positions of the introduced mutations were counted from the 5′-ends of the star and guide strands, respectively: mutations affecting the miRNA’s star strand were denoted with a *, mutations affecting a miRNA’s guide strand with a ^g^.

Unlike siRNAs, miRNAs usually contain mismatched nucleotides at different positions. Considering previous findings [39], we assumed these mismatches to be potential discriminators of the binding to p19. To test this systematically, we first generated an *At*miR162 variant, *At*miR162-0, which, in comparison with the wt *At*miR162, contains two nucleotide changes at positions 4 and 9 in the RNA’s star strand. In this way, a miRNA became available as starting material in which 13 nt of the star and guide strands were fully complementary and capable of forming Watson–Crick base pairs at one terminus (Figure 2A). Binding experiments with p19 determined a similar dissociation constant (K_D_) of *At*miR162-0 as the wt *At*miR162 (i.e., in the range of 0.1–0.2 nM; Figure 2B; Table 1), indicating that *At*miR162-0 was suitable for the planned mutagenesis experiments. Based on *At*miR162-0, we generated 10 further variants by nucleotide exchanges of each of the 5′-terminal 10 positions of the star strand, leaving the guide strand unchanged (see three examples in Figure 2A). The chosen mutagenesis strategy followed a reported strategy [40]; i.e., when a pyrimidine base was present in the guide strand, the complementary nucleotide in the star strand was changed to cytosine; when a purine was in the guide strand, the star strand was changed to the respective other purine (such that A was opposed by G and vice versa). As shown in Figure 2B, binding assays with *At*miR162-0 and each of the mutants revealed that mismatches involving positions 6*, 7* and 10* had evident negative effects on the binding of these RNAs by p19 (see also summary of K_D_ and K_rel_ values in Table 1). Conversely, mismatches at positions 4* and 9* that are also found in the wt *At*miR162 (see also Figure 1) had no effect (Figure 2B). This was congruent with the earlier observation (Figure 2A) that *At*miR162-0 and wt *At*miR162 had similar affinities to p19.

A prominent feature that distinguishes miR168 (binding at low affinity to p19) from miR162 (binding at high affinity to p19) is the presence of G–U base pairs in miR168 (indicated by dots; see, for example, Figure 1). A G–U base pair only involves two hydrogen bonds formed between the nucleobases and shows distinctive structural, chemical and thermodynamic properties in comparison to Watson–Crick base pairs [41]. To determine whether these ‘G–U wobbles’ affect the binding affinity of a miRNA to p19, we next generated variants of *At*miR162-0 in which each of the ten base pairs in the double-stranded region was progressively replaced by G–U. This was performed such that the corresponding nucleotides in the star strand were in each case modified to guanosine, while the nucleotides in the guide strand were modified to uracil (see examples in Figure 3A). Interestingly, in binding assays we observed substantial negative effects on p19-binding with *At*miR162-0 variants that contained G–U-base pairs at positions 6*/14^g^, 7*/13^g^, 8*/12^g^ and 10*/10^g^. The G–U wobbles at positions 7*/13^g^ and 8*/12^g^ inhibited p19 binding most effectively (Figure 3B, Table 1). Given that mispairing at position 8* had no effect (Figure 2B), we took these data as the first indication that a base pair G–U in a terminus of the miRNA specifically attenuated binding of the p19 dimer.

To evaluate this further, we tested other *At*miR162-0 variants with introduced U–G base pairs (inverse G–U) at positions 6*/14^g^, 7*/13^g^ and 8*/12^g^ and additional mismatch mutations at positions 8*/12^g^ and 10*/10^g^ (Figure 3C). Binding assays revealed that G–U or inverse U–G base pairs at position 6*/14^g^ only had a minor effect on the binding of p19. This was clearly different with 7*/13^g^ (Figure 3C, Table 1) where we observed a significant drop in binding, suggesting that at this position all types of changes that affect a perfect complementarity of the RNA strands have a negative effect on binding of p19. Most strikingly, with 8*/12^g^, the binding of p19 to *At*miR162-0 was weakened exclusively by a G–U base pair but not by a U–G base pair introduced at this position (Figure 3C, Table 1). At position 10*/10^g^, where a C–C mismatch had the most negative impact on the binding constant with p19 (see above), an introduced G–U base pair only had a minor impact. Notably, when we compared the isoforms of miR168 of *A. thaliana* and of *N. benthamiana*, it was obvious that all miR-isoforms contain a G–U at 7*/13^g^ and 8^g^/12*, i.e., at similar positions in both the termini of the RNA duplex (Figure 4; see also Section 4 and Appendix A). Accordingly, we considered these observations to be a strong indication that non-canonical base pairing at these positions strongly affects the binding affinity of a miRNA to p19.

Next, we took the reverse approach to *At*miR168a, attempting to modify this miRNA into a stronger p19 binder. Six miR168 variants were tested. Variants 1–3 contained point mutations, which changed U to C at positions 7* and 12* in the star strand and thus converted the G–U pairs of miR168 individually or in combination into G–C base pairs (Figure 5A). In variants 4 and 5, we exchanged additional nucleotides in the star strand that replaced the natural mismatches and one G–U base pair with canonical base pairs. In variant 6, all three G–U base pairs were substituted by canonical base pairs (Figure 5A). Accordingly, variants 4–6 represented siRNA-like variants of miR168 that either contained G–U wobbles (variants 4 and 5) or did not (variant 6). In the binding assay summarized in Figure 5B, solely the *At*miR168 variants 2, 3, 5 and 6 showed a significant increase in the binding affinity to p19 (decrease in the measured K_D_; see also Table 1). Most interestingly, the major common feature of these variants is the modification of the G–U base pair at position 8 of one terminus of the duplex, which is most obvious with miR168 variant 2 (Figure 5).

Thus, together with the earlier observations for the miR162 variants, these data suggested that a G–U base pair located at position 8 in one terminus of a miRNA (position 12 in the complementary strand) represents a specific molecular feature that determines a less stringent binding of the RNA duplex to p19.

Defined amino acid residues control the interaction of p19 with miR162 and miR168. The earlier reported structural analyses of a *CIRV* p19/siRNA complex [22] revealed that two amino acid residues of the protein, K67 and Q107, directly interact with the phosphate backbone of the bound RNA: K67 was indicated to contact bp 8/12, Q107 to contact bp 7/13 (see Figure 6 and Section 3). In addition, we hypothesized that mainly polar contacts of amino acids with the nucleobases were responsible for the differential binding of miRNAs by p19. Thus, we substituted Q6, N8, K67 as well as N106, Q107, V108 and G109 individually by alanine. Furthermore, we performed a T111A exchange, because T111 was earlier suggested to be an important determinant of miRNA binding [39,42]. As a first approach, the mutant *CIRV* p19 proteins were generated by in vitro translation (Appendix A) and tested by EMSA for the binding of *At*miR162 and *At*miR168a, respectively (Figure 7A; see Section 4 for details). Interestingly, five of the nine tested amino acid variants, namely K67A, Q107A, V108A, G109A and T111A, showed apparent defects. While K67A resulted in a general drop in the RNA binding affinity of p19, the variants Q107A, V108A, G109A and T111A showed a significant reduction in binding to miR168 (Figure 7A,B). Since Q107A, V108A and G109A showed the most significant effect on miR168 binding, we next repeated these experiments with the heterologously (*E. coli*) expressed and purified mutant proteins. CD spectroscopy revealed that the respective amino acid exchanges did not cause considerable perturbations to the overall structures of the protein variants (Appendix A). Moreover, we confirmed and extended our earlier findings when we performed EMSA: i.e., in comparison with the wt p19, the Q107A, V108A and G109A variants showed a measurable reduction in the affinity to the applied control-siRNA and to miR162 (Figure 7C). However, with miR168, the affinities of the mutant proteins were so low that K_D_ values could not be assessed by EMSA but only estimated to be in the µM range. Hence, considering the ratios of the binding constants of miR162 or gf698 versus miR168, these were considerably lower in the mutant proteins.

In conclusion, these data suggested that a defined constellation of amino acids of p19, involving particularly amino acid residues 107–109, determines the binding behavior of the VSR to the miRNAs, in particular to miR168.

An unrelated VSR shows similar binding behavior to miR168 variants as p19. In our previous study [36], we also investigated the functional interplay of the VSR 2b of *Tomato Aspermy virus* (*TAV*, genus *Cucumovirus*, family *Bromoviridae*) with different sRNAs, including miRs 162 and 168. Interestingly, we found that despite the fact that both VSRs are unrelated (they consist of completely different primary amino acid sequences) and also have completely different RNA binding modes, p19 and 2b show a very similar miRNA binding profile [36]. Following this idea, we performed here also a binding experiment with *TAV* 2b. As shown in Table 2, similar to p19, the binding affinity of *TAV* 2b and miR168 is low. However, it is significantly increased when the G–U wobble at position 8/12 is modified to a G–C base pair (variant 2 in Figure 5).

These data suggest that different VSRs appear to follow similar rules in binding certain miRNAs.

## 3. Discussion

Virus-encoded VSRs effectively interfere with RNA silencing, for example, by the sequestration of virus-derived and antivirally acting vsiRNAs. Experimental hints by others and ourselves indicated that an additional feature of these proteins involves the inhibition of miRNA function. This can occur in a similar way as with siRNAs, namely by binding with high-affinity (‘highjacking’) and thus removing the RNAs from their cellular activities, in particular the RNA silencing process [36,44,45,46,47,48,49].

A well-known example in this context is miR162, which is efficiently sequestered by p19 and other VSRs (Figure 1). VSR-mediated lowering of the miR162 level is thought to have several effects that support viral infection. On the one hand, it increases mRNA stability and with this the expression level of DCL1, the general pri- and pre-miRNA processor in plant cells [26]. This, in turn, should lead to an increase in the delivery of all types of miRNAs, including those that counteract antiviral silencing, such as miR168 (see below). On the other hand, it has been shown [50] that low miR162/high DCL1 levels, as part of a tight homeostatic DCL interaction network, correlate with lower expression levels of antivirally active DCL4 and thus may represent an additional measure to counteract RNA silencing in the early stages of infection.

Another intensively studied process in virus-modulated RNA silencing is the homeostasis of AGO1 [33]. During viral infection, p19 and other VSRs induce transcriptional accumulation of miR168 via mechanisms that are not yet fully elucidated. miR168, in turn, promotes post-transcriptional repression of the synthesis of the antiviral AGO1 protein [32,51]. In this context, it makes biological sense that the affinity of p19 for miR168 is low (Figure 1) and that miR168 bypasses the sequestration of p19. This is also achieved through additional mechanisms such as the presence of isomiR168 variants that differ in length and structure. Interestingly, the generation of these isomiRs is promoted by a conserved motif containing the G–U wobble 7*/13^g^ that allows flexible internal base-pairing [35].

Consistent with these notions, we previously demonstrated in in vitro silencing and *in planta* experiments that p19 indeed affects miRNA-mediated silencing of the mRNAs DCL1 and AGO1 very differently and strictly according to the miRNA binding profile of the protein. Importantly, these p19–miR interactions are observable early in infection, i.e., when vsiRNA levels are still low and viral genomes unpackaged and thus susceptible to silencing [36].

The aim of this study was to define the basic molecular determinants of p19–miRNA interactions. Using mainly highly purified p19 wt and variant proteins, we showed that the quality of VSR protein–miRNA interactions correlates surprisingly stringently with defined aa/nucleotide contacts. As shown in Figure 2, Figure 3, Figure 4 and Figure 5, the effective binding of a miRNA by p19 is determined to a large extent by the interactions between the base pair 8^g^/12* of the RNA duplex and amino acids 107–109 of the protein. Fittingly, amino acids 107–109 are highly conserved in the p19 proteins of nearly all Tombusviruses [52].

In the available crystal structures, which, however, were obtained with p19/siRNA complexes, the peptide main chain amide of p19 G109 was suggested to be in water-mediated contact with the 2-amine of a guanine at position 8 of the bound siRNA (Figure 6). That this contact is critical for high or low affinity of miRNA binding was most evident when we modified miR162 to a ‘higher K_D_ binder’ and miR168 into a ‘lower K_D_ binder’. That is, single site mutations that converted a classic G–C Watson–Crick base pair at position 8^g^/12* to a G–U wobble (with miR162, Figure 3) or vice versa (with miR168, Figure 5), sufficed to largely disturb or improve the miRNA–p19 interactions, respectively (see also Table 1).

To our knowledge, the G–U wobble is described here for the first time as a prominent determinant of miRNA–protein interactions. G–U wobbles are known to lead to sequence-dependent distortions of an RNA double helix [41]. Specifically in the case of miR168, this appears to be a major reason for the attenuation of the interaction with p19. While in the case of p19 binding to an siRNA G109 can interact with a nucleobase at position 8 (Figure 6), this is apparently not possible for a miRNA containing a G–U base pair that is specifically aligned as 8^g^/12*. Interestingly, all *At* and *Nb* isoforms of miR168 contain a G–U at 7*/13^g^ and 8^g^/12*, i.e., at similar positions in both the termini of the miRNA duplex (see Figure 4 and Appendix A). The G–U wobble at position 8^g^/12* of miR168 is highly conserved in all plant organisms that have been recorded so far [53]. This supports the view that this very position represents a crucial hub of adaptation of the plant to different environmental conditions such as viral infections.

A very interesting observation was that two completely different VSRs, namely Tombusvirus p19 and Cucumovirus 2b, which exhibit entirely different RNA binding modes, nevertheless show the same binding pattern with different miRNAs and also with different mutants of miR168 (Table 2). Thus, adaptation of VSR binding affinities to achieve differential effects on host miRNA activities appears to be a conserved viral mechanism that coordinately modulates cellular gene expression and the antiviral immune response. On the other hand, plants could counteract this property of VSRs by a modified set of miRNAs: single mutations at specific miRNA positions, such as 7/13 or 8/12 of miR168, could provide a higher level of resistance to viral infection through the higher levels of AGO1 produced.

Thus, our study supports the view that the evolution of miRNAs and viral suppressors of RNA silencing is a critical component of the coevolution of plants and plant viruses.

## 4. Materials and Methods

### 4.1. Plasmid Constructs

Cloning of the *CIRV* p19-ORF into the pGEX-6P-1 vector (GE Healthcare, Chalfont St. Giles, UK) and of the *TAV* 2b ORF into the pET SUMO vector (Life Technologies, Carlsbad, CA, USA) for protein synthesis in *E. coli* was described previously [36]. To generate the *CIRV* p19 alanine variants, site directed mutagenesis was performed using a standard protocol (see Appendix A for primers). Successful cloning was verified via sequencing (Seqlab, Goettingen, Germany).

### 4.2. Protein Expression and Purification

GST fusion proteins of the wild-type (wt) and mutant p19 proteins (see text) were expressed in *E. coli* BL21 (DE3) RIPL cells after induction with 1 mM isopropyl-1-thio-β-D-galactopyranoside (IPTG, Roth). For cell lysis, bacterial pellets were re-suspended in lysis buffer 1 (50 mM Tris/Cl pH 7.5, 100 mM NaCl, 1 mM DTT), pre-treated at 4 °C with lysozyme (1.5 mg/g cells, Sigma-Aldrich, Taufkirchen, Germany) and lysed via French press. For purification of fusion proteins, the soluble fraction was loaded onto a GSTrap-column (5 mL, GE Healthcare). After washing with phosphate-buffered saline, the column was equilibrated with cleavage buffer (50 mM Tris/Cl pH 7.5, 100 mM NaCl, 1 mM DTT) and cleavage performed with 100 U of PreScission protease (GE Healthcare) at 4 °C overnight. Following elution with cleavage buffer, the p19 variants were further purified via a Resource Q anion exchange column (GE Healthcare), concentrated to ~40 µM, transferred to storage buffer (10 mM Tris/Cl pH 7.5, 150 mM NaCl, 1 mM EDTA, 1 mM DTT, 50% (*v*/*v*) glycerol) using ultrafiltration, and stored at −20 °C. Protein purity and folding were determined by UV and CD spectroscopy as described previously [36,54]. Dimer formation was confirmed by analytical ultracentrifugation at protein concentrations of 1.2–13 µM (*CIRV* p19) as also described previously [36].

His-SUMO-*TAV*2b fusion protein was expressed in *E. coli* BL21 (DE3) RIPL cells that were grown at 37 °C and induced by 0.4 mM IPTG. After induction, the cells were grown at 25 °C overnight. For lysis, the cells were re-suspended in lysis buffer 2 (50 mM potassium phosphate pH 7.6, 500 mM NaCl, 1 mM DTT containing 1 mM PMSF and protease inhibitor, Roche Diagnostics, Mannheim, Germany), pre-treated at 4 °C with lysozyme (1.5 mg/g cells, Sigma-Aldrich) and lysed via *French press*. The lysate was centrifuged at 12,500× *g* and 10 °C and the soluble fraction containing the *TAV* 2b fusion protein directly loaded onto a HisTrap^TM^ FF column (5 mL, GE Healthcare). After washing with buffer 2, the protein was eluted with buffer 3 (50 mM sodium phosphate pH 7.6, 500 mM NaCl, 500 mM imidazole, 1 mM DTT) and dialyzed against buffer 4 (10 mM Tris/Cl pH 7.6, 25 mM NaCl, 1 mM DTT). The solution containing the fusion protein was applied to an anion exchange chromatography (Hi Prep™Q HP 16/10, GE Healthcare). Elution of the target protein was performed using a linear gradient of buffer 5 (10 mM Tris/Cl pH 7.6, 250 mM NaCl, 1 mM DTT). The His-SUMO tag was then removed by SUMO protease cleavage. Afterwards, the protein was directly subjected to heparin affinity chromatography (2 × 5 mL HiTrap Heparin HP, GE Healthcare), washed with buffer 6 (50 mM Tris/Cl pH 7.6, 100 mM NaCl, 1 mM TCEP) and eluted with a linear gradient of buffer 7 (50 mM Tris/Cl pH 7.6, 1 M NaCl, 1 mM TCEP). Fractions containing the target protein were collected and concentrated using ultrafiltration (Vivaspin^®^ 20, Sartorius, Goettingen, Germany) with molecular weight cut-off 5000 Da. The concentrated protein solution was supplied to a final size exclusion chromatography on a G25 column for removal of impurities, and re-buffering with buffer 6. Final concentration was determined as 2.67 mg/mL. Samples were quickly frozen and stored at −80 °C.

### 4.3. Origin, Generation and Modification of Examined Small RNAs (sRNAs)

The gf698 siRNA used as a control in this study was earlier described by Iki et al., 2010 [55]. The sequences of the applied miRNAs from *Arabidopsis thaliana* (*At*) were derived from the miRBase database (www.mirbase.org (accessed on 2 November 2015)). RNA oligonucleotides (Appendix A) were purchased from biomers (Ulm, Germany). Since phosphorylation was earlier shown to increase the binding affinity of sRNAs by *CIRV* p19 [22], all oligonucleotides were 5′-phosphorylated using T4 polynucleotide kinase (Thermo Fisher Scientific, Waltham, MA) and standard procedures. For radiolabeling, 25 µCi of [γ-^32^P] ATP (3000 Ci/mmol, Hartmann Analytic, Braunschweig, Germany) was added to the phosphorylation reaction. For sRNA annealing, the phosphorylation reactions were stopped by the addition of 25 mM EDTA and reactions of two complementary oligonucleotides combined. After heating at 94 °C for 3 min, the mixture was cooled in steps of 3 °C (for 3 min each) to 25 °C. Hybridized RNA duplexes were then purified with illustra^TM^ microSpin^TM^ G-25 columns (GE Healthcare) as suggested by the manufacturer. The non-mutated miR^g^/miR* *At* and *Nicotiana benthamiana* (*Nb*) duplexes are listed in Appendix A. The mutated miR^g^/miR* *At* duplex variants are listed in Appendix A.

### 4.4. Determination of Binding Affinities

For direct measurement of the binding affinities, radiolabeled siRNA (≤3 pM) was incubated with different concentrations of the purified *CIRV* p19 in binding buffer (20 mM Tris/Cl pH 7.5, 100 mM NaCl, 1 mM EDTA, 1 mM DTT, 0.02% Tween-20) at 24 °C for 1–2 h. For competition experiments with *CIRV* p19, the unlabeled but phosphorylated miRNAs were titrated against radiolabeled gf698 siRNA (≤3 pM) that had been bound to a fixed concentration of protein (0.5 nM *CIRV* p19 dimer) in binding buffer overnight. The samples were mixed at 0.25 *v*/*v* with loading dye (50 mM Tris/Cl pH 7.5, 10 mM EDTA, 0.002% bromophenol blue, 0.002% xylene cyanol, 50% *v*/*v* glycerol), analyzed by PAGE on a native 6% Tris-Borate EDTA (TBE) gel. The protein-bound and free RNAs were detected by phosphorimaging (Storm 860, Molecular Dynamics, Caesarea, Israel) and quantified by ImageQuant software (GE Healthcare). Since *CIRV* p19 always forms a dimer (see Figure 6), the fractions of p19-bound radiolabeled gf698 were plotted versus the free protein dimer concentrations and fitted to a rectangular hyperbola for binding constant determination in direct binding reactions.
(1)$$S_{n}=\frac{D_{t}{}^{n}}{K_{d}{+D}_{t}{}^{n}}$$

$S_{n}$ is the bound sRNA fraction, D_t_ is the total protein dimer concentration, K_d_ is the binding constant, n is the cooperativity.

For competition reactions that were performed as earlier described [36] with purified *TAV* 2b, the gf698 siRNA and the respective miRs, the fractions of bound radiolabeled gf698 were plotted versus free competitor RNA concentrations and fitted to the following equation:(2)$$S_{n}=\frac{1}{2\cdot R_{t}}\cdot \left[K_{d}+\frac{K_{d}}{K_{c}}\cdot C_{t}{+P}_{t}{+R}_{t}-\sqrt{{\left(K_{d}+\frac{K_{d}}{K_{c}}\cdot C_{t}{+P}_{t}{+R}_{t}\right)}^{2}-4\cdot R_{t}\cdot P_{t}}\right]$$

R_t_ is the total concentration of radiolabeled gf698, C_t_ is the total competitor RNA concentration, P_t_ is the total protein concentration, K_c_ is the apparent dissociation constant of the competitor RNA. All measurements were done at least in triplicate (see indications in the text and figure legends). Data were expressed as the mean and standard deviations. Student’s *t*-test was performed with mean values.

### 4.5. Cell Culture and Preparation of BYL

*Nicotiana tabacum* BY2 cells were cultured at 23 °C in Murashige–Skoog liquid medium. Evacuolated BY2 protoplasts were obtained by percoll gradient centrifugation and cytoplasmic extract (BYL) prepared as described earlier [56,57].

### 4.6. In Vitro Translation of p19 Variants

The in vitro translation of p19 variants was performed in 50% (*v*/*v*) BYL at the previously described conditions [57]. Briefly, 0.5 µg of in vitro transcribed p19 RNA was translated in a 20 µL reaction in BYL for 60 min at 25 °C. For the detection of the translation products, 10 µCi of L-[^35^S]-methionine (1000 Ci/mmol, Hartmann Analytic) was added to the reaction. The samples were separated by 15% (*w*/*v*) SDS-PAGE and the labeled proteins visualized by phosphorimaging (Storm 860, Molecular Dynamics). For initial screening of miRNA interactions, ca. 30 pmol of radiolabeled *At*miR162 or *At*miR168a was added to the translation reaction and incubated for 60 min at 25 °C. The samples were mixed with 0.25 *v*/*v* of loading dye and analyzed by PAGE on a 6% TBE gel under non-denaturing conditions. Bands corresponding to bound and free radiolabeled RNA were detected by phosphorimaging (Storm 860, Molecular Dynamics) and quantified by ImageQuant software (GE Healthcare). All measurements were done at least in triplicate. Data were expressed as the mean and standard deviation, and the results were statistically analyzed using Student’s *t*-test.

## Figures and Tables

**Figure 1 ijms-23-04977-f001:**
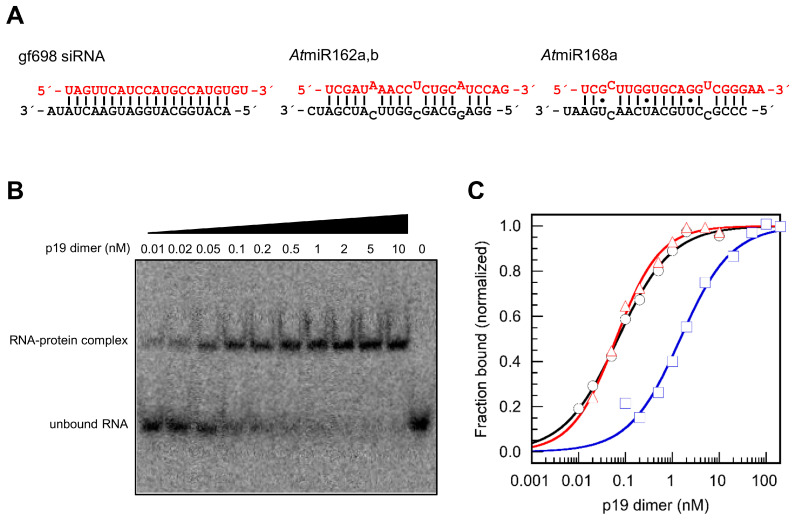
**p19 binds different small RNAs at different affinities.** (**A**) Schematic representations of the tested small RNAs gf698 (siRNA), *At*miR162 and *At*miR168a. Guide (^g^) strands are represented in red. Dots indicate G–U wobbles; mismatched nucleotides are set outwards. (**B**) Representative gel image of an electrophoretic mobility shift assay (EMSA) showing a direct binding experiment that was performed with 5′-labeled *At*miR162 (≤3 pM) and the indicated amounts of *CIRV* p19. (**C**) Representative analyses of binding equilibrium data of *CIRV* p19 bound to gf698 siRNA (black circles), *At*miR162 (red triangles) and *At*miR168a (blue squares), respectively. Data were fitted to a rectangular hyperbola with cooperativity, Formula (1) [36]. Results of the fits are plotted as lines.

**Figure 2 ijms-23-04977-f002:**
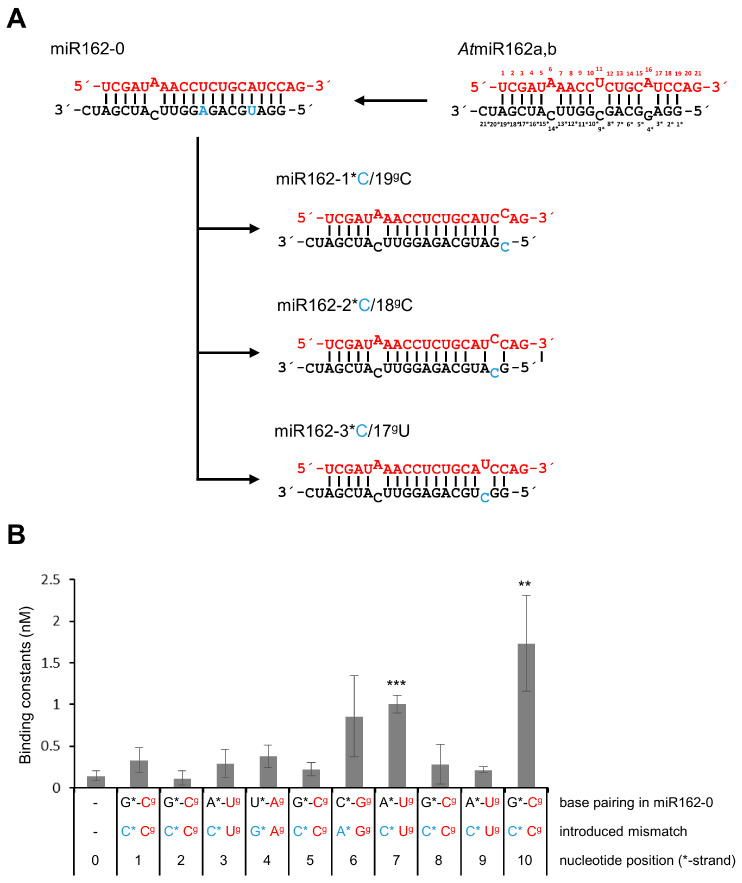
Mismatches affect the interactions of miR162 with p19. (**A**) Work scheme showing the generation of miR162-0 and of three of ten mutant miR162-0 forms, in which nucleotides involving positions 1* to 10* were stepwise exchanged. (**B**) Results of binding assays that were determined with *At*miR162-0 and each of the mutated variants containing mismatches (*n* = 3–4). Given are the binding constants K_D_ values that were measured as described in Materials and Methods (Formula (1)). In all figures, exchanged nucleotides are indicated in blue. Error bars specify standard deviations. Asterisks indicate statistically significant differences in comparison to *At*miR162-0 (** *p* < 0.01; *** *p* < 0.001).

**Figure 3 ijms-23-04977-f003:**
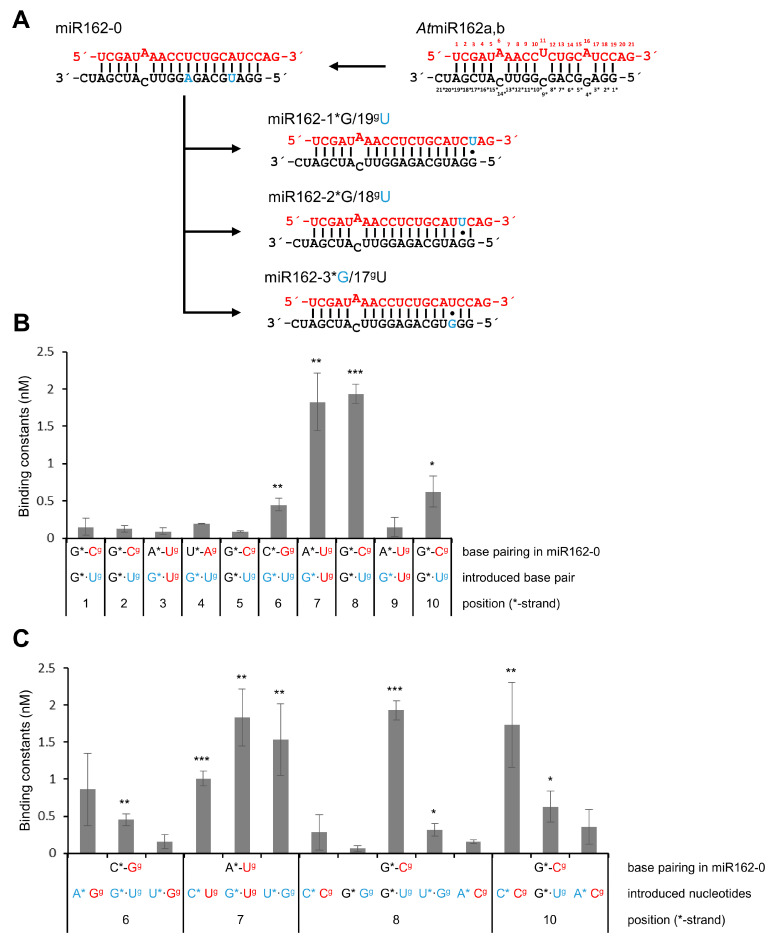
G–U wobbles affect the interactions of miR162 with p19. (**A**) Work scheme describing the stepwise introduction of G–U wobbles into the *At*miR162-0 duplex with three of ten examples (see text). (**B**) Binding assays (performed as described in Figure 2) with each of the mutated variants containing G–U wobbles (*n* = 3–5). (**C**) Binding assays (performed as described in Figure 2) with *At*miR162 variants that contained different alterations at positions 6*/14^g^, 7*/13^g^, 8*/12^g^, and 10*/10^g^, respectively (*n* = 3–5). See text for details. In all figures, exchanged nucleotides are indicated in blue. Error bars specify standard deviations. Asterisks indicate statistically significant differences in comparison to *At*miR162-0 (see also Figure 2; * *p* < 0.05; ** *p* < 0.01; *** *p* < 0.001).

**Figure 4 ijms-23-04977-f004:**
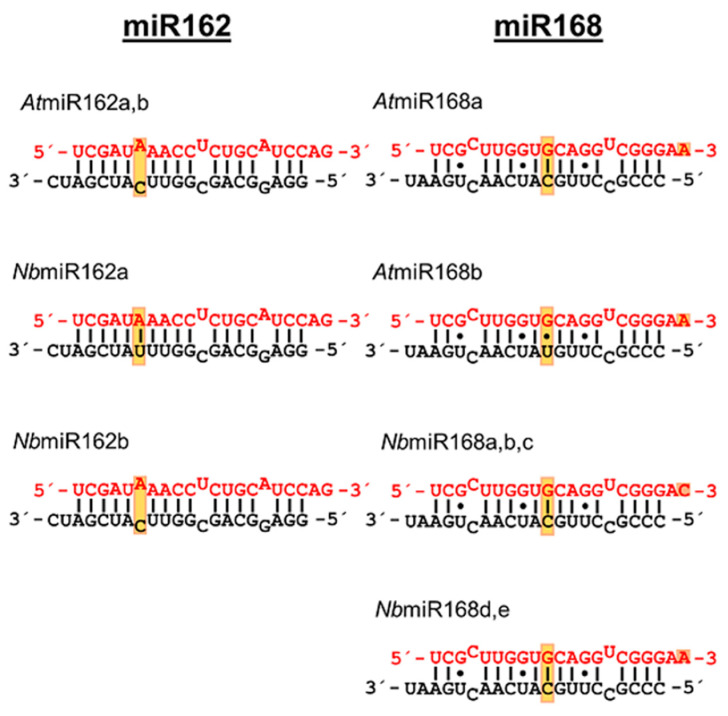
**Structure of *At* and *Nb* miR162 and miR168 isoforms.** Schematic representations of different miR162 and miR168 isoforms. The miRNAs from *At* are annotated in miRBase. The shown miRNAs from *Nb* derived from a BLAST-search of micro RNA precursor sequences versus the *Nb* draft genome at www.solgenomics.net (accessed on 1 November 2015). Guide and star strands as well as G–U wobbles and mismatched nucleotides are indicated as in Figure 1. Positions within a miR family with sequence variations are boxed.

**Figure 5 ijms-23-04977-f005:**
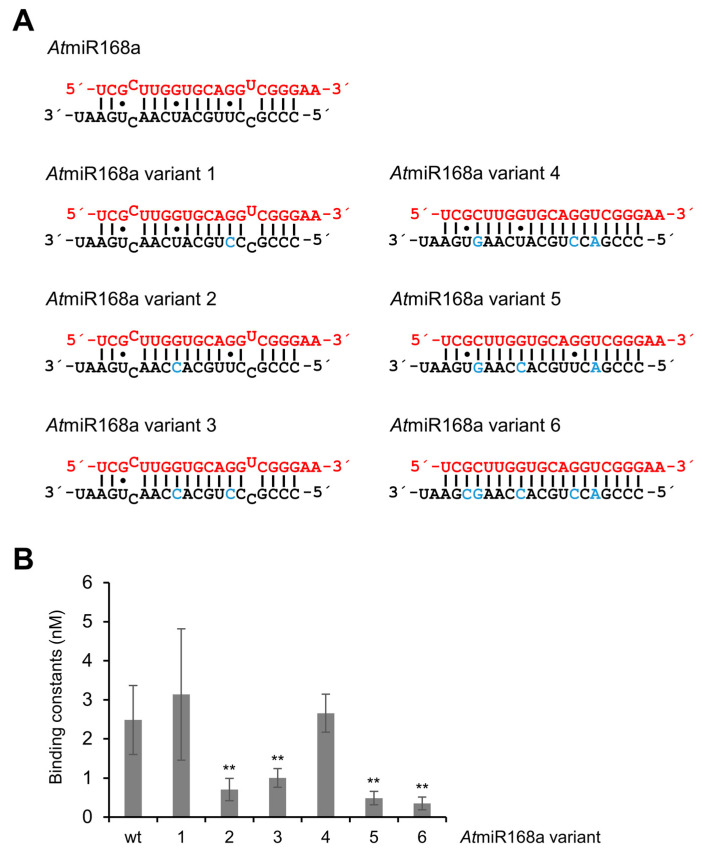
Substitution of a defined G–U wobble at position 8^g^/12* in miR168 increases the affinity of the RNA to p19. Six variants of *At*miR168a (*Nb*miR168d,e), schematically depicted in (**A**) were generated and tested for binding (**B**) as described in Figure 2 and in the text (*n* = 3–4). Exchanged nucleotides are indicated in blue. Error bars specify standard deviations. Asterisks indicate statistically significant differences in comparison to *At*miR168a (** *p* < 0.01).

**Figure 6 ijms-23-04977-f006:**
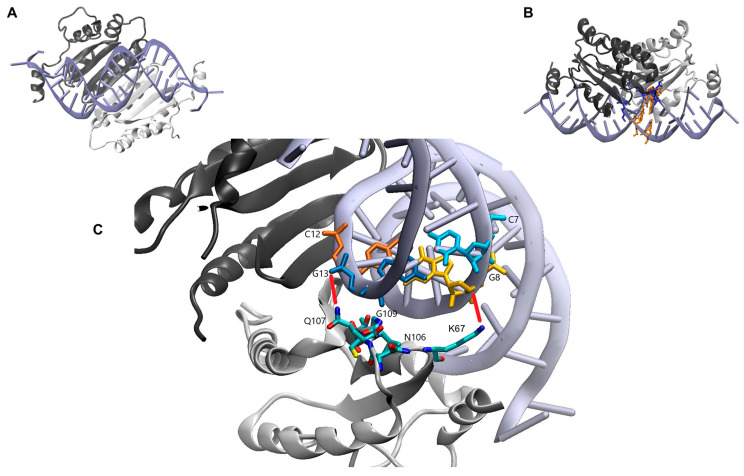
Published crystal structure of *CIRV* p19 in complex with a siRNA (PDB-Code: 1rpu). (**A**) Crystal structure of the *CIRV* p19 dimer in complex with a siRNA published by Vargason and colleagues [22]. The siRNA is colored in purple, the protein monomers of the dimer in dark and light grey, respectively. (**B**) Same view as in (**A**) rotated by 90°. Base pairs C7/G13 and G8/C12 are highlighted in orange; amino acid residues of one monomer subunit (dark grey) of the p19 dimer positioned to these nucleotides are highlighted in blue. (**C**) Zoom-in of (**B**). The base pair C7/G13 in the siRNA is labeled in light blue/dark blue, the base pair G8/C12 in yellow/orange, respectively. The amino acids K67, N106, Q107, V108, G109 and T111 of one monomer (light grey) within the dimer are displayed. Red lines indicate interactions between the p19 protein and the phosphate backbone. All schemes were generated using the program VMD [43], considering the dimer of *CIRV* p19 as the biological entity. Note that the interactions of each monomer within the dimer with the double-stranded nucleic acid are similar.

**Figure 7 ijms-23-04977-f007:**
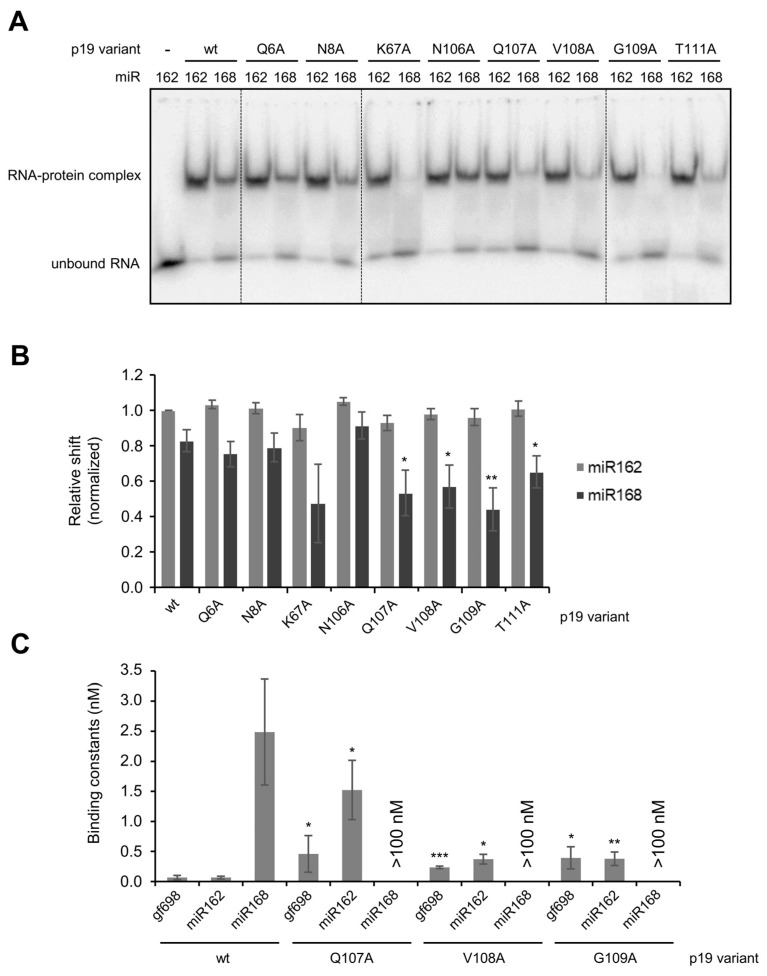
Defined mutants of p19 show altered miR162/miR168 complex stabilities. (**A**) The mutant p19 proteins were in vitro translated in BYL (see also Appendix A) and complex formation with miR162 and miR168 was tested by EMSA. A representative gel is shown. (**B**) Shift intensities of the labeled RNAs (relative shifts with respect to total signal intensities) were determined (*n* = 3). Error bars specify standard deviations. (**C**) Mutant proteins Q107A, V108A and G109A, which revealed the most significant negative effects on RNA–protein binding in (**B**) were purified and the binding constants (K_D_ values) determined with the indicated small RNAs (*n* = 3–5). Higher ordered complexes formed at protein concentrations ≥ 0.5 µM (not shown) and for that reason, K_D_ values ≥ 100 nM could only be estimated. Error bars specify standard deviations. Asterisks indicate statistically significant differences in comparison to p19 wt (* *p* < 0.05; ** *p* < 0.01; *** *p* < 0.001).

**Table 1 ijms-23-04977-t001:** Binding constants of all tested miR162 variants and miR168 variants in binding to *CIRV* p19.

sRNA Name	Property	K_D_ (nM)	^a^K_rel_
**gf698 (siRNA)**	siRNA control	0.06 ± 0.04	-
** *At* ** **miR162**	Wild-type of *At*miR162	0.06 ± 0.02	1.00 ± 0.31
**miR162-0**	miR162 variant with Watson–Crick base pairing terminus	0.15 ± 0.06	2.31 ± 0.92
**miR162-19^g^C/1*C**	miR162 variant with C/C mismatch at 1*/19^g^	0.34 ± 0.15	5.24 ± 2.31
**miR162-18^g^C/2*C**	miR162 variant with C/C mismatch at 2*/18^g^	0.12 ± 0.08	1.85 ± 1.23
**miR162-17^g^U/3*C**	miR162 variant with C/U mismatch at 3*/17^g^	0.30 ± 0.17	4.62 ± 2.62
**miR162-16^g^A/4*G**	miR162 variant with G/A mismatch at 4*/16^g^	0.38 ± 0.13	5.86 ± 2.00
**miR162-15^g^C/5*C**	miR162 variant with C/C mismatch at 5*/15^g^	0.23 ± 0.08	3.55 ± 1.23
**miR162-14^g^G/6*A**	miR162 variant with A/G mismatch at 6*/14^g^	0.86 ± 0.49	13.26 ± 7.55
**miR162-13^g^U/7*C**	miR162 variant with C/U mismatch at 7*/13^g^	1.01 ± 0.10	15.57 ± 1.54
**miR162-12^g^C/8*C**	miR162 variant with C/C mismatch at 8*/12^g^	0.28 ± 0.24	4.32 ± 3.70
**miR162-11^g^U/9*C**	miR162 variant with C/U mismatch at 9*/11^g^	0.22 ± 0.03	3.39 ± 0.46
**miR162-10^g^C/10*C**	miR162 variant with C/C mismatch at 10*/10^g^	1.73 ± 0.57	26.67 ± 8.79
**miR162-19^g^U/1*G**	miR162 variant with G•U wobble at 1*/19^g^	0.15 ± 0.12	2.31 ± 1.85
**miR162-18^g^U/2*G**	miR162 variant with G•U wobble at 2*/18^g^	0.13 ± 0.03	2.00 ± 0.31
**miR162-17^g^U/3*G**	miR162 variant with G•U wobble at 3*/17^g^	0.09 ± 0.04	1.39 ± 0.62
**miR162-16^g^U/4*G**	miR162 variant with G•U wobble at 4*/16^g^	0.17 ± 0.05	2.62 ± 0.77
**miR162-15^g^U/5*G**	miR162 variant with G•U wobble at 5*/15^g^	0.10 ± 0.01	1.54 ± 0.15
**miR162-14^g^U/6*G**	miR162 variant with G•U wobble at 6*/14^g^	0.45 ± 0.08	6.94 ± 1.23
**miR162-13^g^U/7*G**	miR162 variant with G•U wobble at 7*/13^g^	1.82 ± 0.38	28.05 ± 5.86
**miR162-12^g^U/8*G**	miR162 variant with G•U wobble at 8*/12^g^	1.93 ± 0.13	29.75 ± 2.00
**miR162-11^g^U/9*G**	miR162 variant with G•U wobble at 9*/11^g^	0.14 ± 0.12	2.16 ± 1.85
**miR162-10^g^U/10*G**	miR162 variant with G•U wobble at 10*/10^g^	0.63 ± 0.21	9.71 ± 3.24
**miR162-14^g^G/6*U**	miR162 variant with U•G wobble at 6*/14^g^	0.15 ± 0.09	2.31 ± 1.39
**miR162-13^g^G/7*U**	miR162 variant with U•G wobble at 7*/13^g^	1.53 ± 0.49	23.58 ± 7.55
**miR162-12^g^G/8*U**	miR162 variant with U•G wobble at 8*/12^g^	0.31 ± 0.08	4.78 ± 1.23
**miR162-12^g^C/8*A**	miR162 variant with A/C mismatch at 8*/12^g^	0.16 ± 0.03	2.47 ± 0.46
**miR162-12^g^G/8*G**	miR162 variant with G/G mismatch at 8*/12^g^	0.07 ± 0.04	1.08 ± 0.62
**miR162-10^g^C/10*A**	miR162 variant with A/C mismatch at 10*/10^g^	0.35 ± 0.24	5.39 ± 3.70
** *At* ** **miR168a**	Wild-type of *At*miR168a	2.49 ± 0.88	1.00 ± 2.22
** *At* ** **miR168a variant 1**	*At*miR168a variant without G•U wobble at 7*/13^g^	3.13 ± 1.67	1.26 ± 2.84
** *At* ** **miR168a variant 2**	*At*miR168a variant without G•U wobble at 8^g^/12*	0.71 ± 0.28	0.29 ± 0.64
** *At* ** **miR168a variant 3**	*At*miR168a variant without wobbles at 8^g^/12* and 17^g^/3*	1.00 ± 0.24	0.40 ± 0.89
** *At* ** **miR168a variant 4**	*At*miR168a variant without G•U wobble at 7*/13^g^ and mismatches	2.65 ± 0.49	1.07 ± 2.34
** *At* ** **miR168a variant 5**	*At*miR168a variant without G•U wobble at 8^g^/12* and mismatches	0.48 ± 0.17	0.19 ± 0.43
** *At* ** **miR168a variant 6**	siRNA variant of *At*miR168a	0.34 ± 0.16	0.14 ± 0.31

^a^K_rel_ denotes the ratio of K_D_ of the respective RNA (variant) to the K_D_ of the wild-type RNA.

**Table 2 ijms-23-04977-t002:** RNA binding behavior of *TAV* 2b.

Competitor RNA	K_C_ (nM)	^a^K_rel_
**gf698 (siRNA)**	0.53 ± 0.08	1.00 ± 0.16
** *At* ** **miR162**	0.67 ± 0.09	1.26 ± 0.18
** *At* ** **miR168a**	12.10 ± 0.70	22.83 ± 1.64
** *At* ** **miR168a variant 2**	1.10 ± 0.20	2.08 ± 0.39
** *At* ** **miR168a variant 3**	2.20 ± 0.40	4.15 ± 0.77
** *At* ** **miR168a variant 5**	1.40 ± 0.20	2.64 ± 0.39
** *At* ** **miR168a variant 6**	0.38 ± 0.05	0.72 ± 0.10

^a^K_rel_ denotes the ratio of K_C_ of the respective RNA (variant) to the K_C_ of gf698 (siRNA). The K_D_ of gf698 to *TAV* 2b was determined to 0.68 ± 0.15. The binding behavior of *TAV* 2b to the listed RNAs (examples) was determined by competitive binding assays as described earlier [36] and in Section 4.

## Data Availability

All data of this study are available and maintained according to the guidelines of the MDPI Research Data Policies and the Deutsche Forschungsgemeinschaft, DFG.

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
