# Peer review of "RNA and Protein Determinants Mediate Differential Binding of miRNAs by a Viral Suppressor of RNA Silencing Thus Modulating Antiviral Immune Responses in Plants"

_ijms, 2022, doi:10.3390/ijms23094977_

Round 1

Reviewer 1 Report

The manuscript “RNA and protein determinants mediate differential binding of miRNAs by a viral suppressor of RNA silencing thus modulating antiviral immune responses in plants” by Pertermann et al. is a nice biochemical work expanding on earlier observations regarding the uneven affinity of p19 and similar VSRs for plant miRNAs, especially those targeting the host RNAi machinery. This differential affinity likely has important biological consequences for both the host and the virus during infection. The authors have undertaken a systematic analysis of miR162 variants to evaluate the contribution of duplex-disturbing mutations on its affinity for p19. They further rationalised their observations by successfully improving the binding of the naturally poor ligand miR168 and by providing interaction data on the protein side. The manuscript is well written, the data are of high quality, and the paper can be recommended for publication. I only have a few minor comments/questions.

  1. The Kds for WT variants of both miRNAs are identical (including SEs) to those reported in reference (36). This is either a striking and unlikely coincidence, or the authors have actually re-used their published data for these two miRNAs. In the latter case, this should be explicitly mentioned in the text, and the authors should not write in ll. 172-175 that they “confirmed earlier data…” In the same vein, Fig. 1C seems to feature the same siRNA curve as in Fig. 2B from reference (36). This should be either acknowledged or replaced with a different, unpublished dataset.
  2. The authors have convincingly demonstrated that mismatches and G:U base pairs in positions 6-8 and 10 may be deleterious for p19 binding. Does the presence/absence of such features correlate with p19 coIP enrichments of miRNAs in their previous study (reference 36, Fig. 1)? This could make a case for a more general conclusion.
  3. Table 2 contains actual data, and I would recommend moving it (and the accompanying text) to the Results section.
  4. Ll. 92-93: convert “40,000 rpm” in g.
  5. L. 116: how much EDTA?
  6. Ll. 124-127, 138-143: unless I am mistaken, this manuscript does not contain competition experiments.
  7. Ll. 135-136: Kd is the dissociation constant; n is typically called ‘Hill coefficient’ and is, indeed, a measure of cooperativity. A curiosity question: the authors decided to use a general Hill equation model. What kind of n (Hill coefficients) did they observe in their experiments?
  8. Ll. 255-259: discussing the data presented in Fig. 7A, B, the authors seem to overstate the effect on miR162, which is really minute and statistically insignificant. I would only keep conclusions about miR168, keeping in mind that such a binding assay is necessarily semi-quantitative and should be interpreted with caution.
  9. L. 259: reference to Table 1 is inappropriate in this context.

Reviewer 2 Report

Pertermann et al studied the interaction between some certain miRNAs and the P19 protein of tombusviruses. Measured the affinity of p19 and miR162 and miR168. Both miRNas are very interesting because mir162 targets DCL1 and mir168 downregulates AGO1, therefore, as the authors pinpointed, the cumulative effect of these miRNAs might play an important role in the outcome of the antiviral RNAi response against the virus. Authors mapped the position and the nature of nucleotides in the miRNA duplex have a high impact on the binding affinity to p19. And importantly they also identified the key amino acids in p19 responsible for binding of these particular miRNAs.

I believe these study is timely interesting and has a high impact on the field, therefore I suggest the acceptance of the MS as it is.

Author Response

We thank the referee for this positive and supportive statement.